# A Highly Efficient Indirect *P. pastoris* Surface Display Method Based on the CL7/Im7 Ultra-High-Affinity System

**DOI:** 10.3390/molecules24081483

**Published:** 2019-04-15

**Authors:** Shuntang Li, Jie Qiao, Siyu Lin, Yi Liu, Lixin Ma

**Affiliations:** 1State Key Laboratory of Biocatalysis and Enzyme Engineering, School of Life Sciences, Hubei University, Wuhan 430062, China; 20160099@hubu.edu.cn (S.L.); babyqiao@hotmail.com (J.Q.); jingzilive@163.com (S.L.); 2Hubei Key Laboratory of Industrial Biotechnology, School of Life Sciences, Hubei University, Wuhan 430062, China; 3Hubei Collaborative Innovation Center for Green Transformation of Bio-Resources, School of Life Sciences, Hubei University, Wuhan 430062, China

**Keywords:** surface display, *P. pastoris*, *E. coli*, Im7, CL7 tag

## Abstract

Cell surface display systems for immobilization of peptides and proteins on the surface of cells have various applications, such as vaccine generation, protein engineering, bio-conversion and bio-adsorption. Though plenty of methods have been established in terms of traditional yeast surface display systems, the development of a universal display method with high efficiency remains a challenge. Here we report an indirect yeast surface display method by anchoring Im7 proteins on the surface of *P. pastoris*, achieving highly efficient display of target proteins, including fluorescence proteins (sfGFP and mCherry) or enzymes (human Arginase I), with a CL7 fusion tag through the ultra-high-affinity interaction between Im7 and CL7. This indirect *P. pastoris* surface display approach is highly efficient and provides a robust platform for displaying biomolecules.

## 1. Introduction

Taking advantage of natural surface proteins or their fragments as the anchoring motifs, peptides or proteins of interest can be displayed on the surfaces of various microorganisms or cells such as phages [1], spores [2], *E. coli* [3], mammalian [4] and yeast cells [5]. Depending on the characteristics of passenger and carrier proteins, C-terminal, N-terminal and sandwich fusion strategies were used for surface display [6]. The surface display peptides or proteins have a plethora of biotechnological and industrial applications, including development of vaccines through displayed antigens to elicit antigen specific immune responses [7], target peptides and proteins screening from libraries using fluorescence-activated cell sorting (FACS) [8], construction of whole-cell biocatalysts, as well as the design of bio-sensor and electro-chemical devices [9]. 

Yeasts are genetically amenable and can be easily grown with low technological threshold [6]. Thus, many kinds of yeast species, especially for *S. cerevisiae*, have been widely used for surface display. Of note, the large cell size of yeast allows surface display of a relatively large number of protein molecules per cell (~10^4^–10^5^). Combined with FACS, researchers can readily achieve quantitation of the surface-display proteins as well as rapid library screening [6]. Other species of yeast and fungi have also been employed for surface display. Among them, the most used yeast is *P. pastoris* which can reach a much higher cell density than *S. cerevisiae* in fermentation, as well as can be utilized for preparation of recombinant protein. Nevertheless, the incumbent yeast display methods, where the target proteins are directly displayed on the cell surface, have many limitations. One of which is the unsubstantial efficiency. An indirect *P. pastoris* cell surface display methodology has therefore been reported [10]. In the study, a short biotin acceptor peptide (BAP) derived from *E. coli* biotin ligase (BirA) was fused to the N-terminus of a yeast surface anchor protein termed Flo428. The BAP-Flo428 fusion protein and BirA were co-expressed, leading to the display of the biotinylated BAP on *P. pastoris* cell surface. Finally, the addition of streptavidin-FITC (fluorescein isothiocyanate) resulted in the display of streptavidin-FITC. 

Recently, Vassylyev et al. reported an efficient one-step ultra-high-affinity chromatography system for the purification of proteins with high yield, high purity and high activity [11]. In this system, the 16 KDa CL7 tag which was engineered from the *E. coli* Colicin E7 DNase (CE7) retains the ultra-high binding affinity (K_D_~10^−14^–10^−17^ M) with its inhibitor 10 KDa Immunity protein 7 (Im7). Inspired by the work, we sought to develop a universal indirect *P. pastoris* surface display method based on the CL7/Im7 ultra-high-affinity system (Figure 1). Accordingly, Im7 was fused to the N-terminus of SED1 protein [12] from *S. cerevisiae* and the fusion protein Im7-SED1 could be efficiently displayed on the cell surface of *P. pastoris*. Based on the fluorometric assay using CL7 fused sfGFP, we found that ~2.8 × 10^6^ of CL7-sfGFP molecules per yeast cell were indirectly displayed on the surface of *P. pastoris*. Furthermore, in a practice study, the enzyme activities of human Arginase I immobilized on the yeast surface by this approach were measured, resulting in ~ 5045 U per 1 g dry yeast cells. Taken together, we developed a highly efficient *P. pastoris* surface display system which has wide application prospects. 

## 2. Results and Discussion 

### 2.1. General Strategy 

To establish a novel indirect *P. pastoris* cell surface display system that can efficiently display unnatural molecules, we take advantage of the ultra-high-affinity interaction of Im7 and CL7 tag in this work. The principle of our method is illustrated in Figure 1. Initially, the gene of Im7-SED1 fusion protein was integrated into *P. pastoris* genomes. Then, after induction of yeast cells by 1% (*v*/*v*) methanol in BMMY, the Im7-SED1 was displayed on the yeast surface. At last, the addition of CL7 fused proteins resulted in the highly efficient indirect display of target proteins on the yeast cell surface via the super strong interaction of Im7 and CL7 partner (K_D_~10^−14^–10^−17^ M).

### 2.2. Plasmid Construction

Firstly, the DNA fragments encoding CL7, sfGFP, mCherry, human Arginase I, CL7-sfGFP, CL7-mCherry and CL7-human Arginase I (CL7-huArg I) were amplified by PCR using corresponding primers. Then, the DNA fragments of CL7-sfGFP, CL7-mCherry and CL7-human Arginase I were cloned into vector pET23a-gfp [13] using *Nde* I and *Xho* I restriction enzymes. In addition, the DNA fragments encoding for a (G4S)2 linker and an HRV 3C recognition site were also inserted between CL7 and the target proteins, resulting in plasmids pET23a-CL7-sfGFP, pET23a-CL7-mCherry and pET23a-CL7-huArg I (Figure 2A), respectively. Next, the yeast surface anchor protein SED1 from *S. cerevisiae* without its signal sequence was amplified [12]. Based on the *P. pastoris* expression vector pPICZαA, we constructed the plasmid pPICZαA-HA-SED1 (Figure 2B) which contains an HA epitope, a multiple clone site, a glycine-serine peptide linker as well as the anchor protein SED1. The restriction enzyme sites *Nhe* I, *Nde* I, *Hpa* I and *Apa* I were inserted as the multiple cloning sites. Similarly, the plasmid pPICZαA-HA-Im7-SED1 was constructed following the workflow shown in Figure 2C.

### 2.3. Expression of CL7 Fusion Proteins

The fusion proteins including CL7-sfGFP, CL7-mCherry and CL7-huArg I were expressed in BL21(DE3) *E. coli* and purified by Ni^2+^-affinity chromatography (Ni-NTA). The protein samples were detected by 12% SDS-PAGE gels (Figure 3) and analyzed by software ImageJ (Table 1). The recovery efficiency of CL7 fusion proteins eluted by different concentrations of imidazole were calculated. Typically, CL7-sfGFP and CL7-mCherry eluted with 200 mM imidazole, as well as CL7-huArg I eluted with 150 mM and 200 mM imidazole were collected and concentrated. All protein concentrations were determined by the Bradford method using a protein assay kit.

### 2.4. Flow Cytometry Analysis

The GS115/pPICZαA, GS115/pPICZαA-HA-SED1 and GS115/pPICZαA-HA-Im7-SED1 cells were induced in BMMY medium containing 1% (*v*/*v*) methanol for 24 h and were then analyzed by flow cytometry [12]. As a negative control, the unlabeled GS115/pPICZαA cells were used (Figure 4A). The GS115/pPICZαA, GS115/pPICZαA HA-SED1 and GS115/pPICZαA HA-Im7-SED1 cells were labeled by DyLight 649-conjungated goat anti-mouse antibodies, as well as by CL7-sfGFP proteins and mouse anti-HA tag monoclonal antibodies. None of DyLight 649 and CL7-sfGFP fluorescence were observed for GS115/pPICZαA cells (Figure 4B), indicating that there was non-specific protein adsorption on the yeast cell surface. The GS115/pPICZαA HA-SED1 cells showed red DyLight 649 fluorescence (Figure 4C), however no green CL7-sfGFP fluorescence (Figure 4C). In contrast, the GS115/pPICZαA HA-Im7-SED1 cells showed both red and green fluorescence (Figure 4D). These results demonstrate that the HA-SED1 and HA-Im7-SED1 fusion proteins were anchored on the yeast cell surface and the cell surface immobilization of CL7-sfGFP proteins was indirectly displayed by Im7 proteins specifically (Figure 4).

### 2.5. Fluorescence Microscopy

The fluorescence microscopy assays (Figure 5) were performed using the above-mentioned *P. pastoris* cells in the presence of CL7-mCherry fusion proteins (Figure 5, right row) and in the presence of mouse anti-HA tag monoclonal antibodies together with FITC-conjugated goat anti-mouse antibodies (Figure 5, middle row). The GS115/pPICZαA cells did not show green and red fluorescence, while the GS115/pPICZαA HA-SED1 cells only showed green fluorescence on the edges. In contrast, both green and red fluorescence were observed on the edges of GS115/pPICZαA HA-Im7-SED1 cells. Collectively, the fluorescence microscopy data and flow cytometry results shown above clearly demonstrate that sfGFP and mCherry were indirectly displayed on the surface of *P. pastoris* cells through the interaction between Im7 and CL7 tags.

### 2.6. Western Blotting and Blue Light Transmitter Analysis

The display of HA-SED1 and HA-SED1-Im7 proteins on the surface of *P. pastoris* cells (Figure 6a, lane 1 and lane 2) were validated by western-blot with HA specific antibodies. As a control, no bands were detected for GS115/pPICZαA cells (Figure 6a, lane 3). The observed molecular weights of HA-SED1 and HA-Im7-SED1 were around 180 kDa, which are both much higher than the anticipated values (36 kDa for HA-SED1 and 45 kDa for HA-Im7-SED1), possibly due to the glycosylation of these SED1 fusion proteins. Under the blue light transmitter (Figure 6b), the GS115/pPICZαA HA-Im7-SED1 cells incubated with CL7-sfGFP showed a very bright green (Figure 6b, tube 4). In contrast, the GS115/pPICZαA HA-SED1 cells in the presence or absence of CL7-sfGFP (Figure 6b, tube 1 and 2), as well as the GS115/pPICZαA HA-Im7-SED1 cells (Figure 6b, tube 3) in the absence of CL7-sfGFP did not show green fluorescence.

### 2.7. Fluorometric Assay

The CL7-sfGFP indirectly displayed on the *P. pastoris* surface were quantified according to the protocol [14]. The excitation wavelength and emission wavelength of CL7-sfGFP in TBS (tris-buffered saline, pH 7.6) were determined by wavelength scanning (Figure 7). Thus, the wavelength of 490 nm and 512 nm were used as the optimal excitation and emission wavelength. To prepare the calibration curve of CL7-sfGFP (Figure 8), the fluorescence intensities of distinct amounts of CL7-sfGFP fusion proteins in 1 mL TBS were used. Next, the fluorescence intensity and OD_600_ of cells incubated with different CL7-sfGFP were examined and the fluorescence intensity/OD_600_ (F/A) values were calculated and are summarized in Figure 9. The GS115/pPICZαA-HA-SED1 cells incubated with 120 μg of CL7-sfGFP showed a similar F/A value (Table 2) in comparison to GS115/pPICZαA-HA-SED1 cells without CL7-sfGFP proteins (*p* > 0.05), indicating that there is no non-specific binding between CL7-sfGFP and *P. pastoris* cells or surface displayed HA-SED1. On the other hand, the F/A values of GS115/pPICZαA-HA-Im7-SED1 cells incubated with 60 μg and 100 μg of CL7-sfGFP proteins showed no obvious differences (*p* > 0.05), which indicates that 60 μg CL7-sfGFP already saturated surface displayed Im7 proteins. The calculated CL7-sfGFP displayed on yeast surface is ~10.4 μg per mL of 1 OD_600_
*P. pastoris* cells. Paus et al. indicated that 1 mL of *P. pastoris* (OD_600_ = 1) has ~5 × 10^7^ cells [15]. Therefore, the number of indirect displayed CL7-sfGFP molecules per cell should be calculated as N = m * NA/(M*5 × 10^7^), where m is 10.0 × 10^−6^ g, M is molecule weight of CL7-sfGFP (44.4 kDa) and NA is the Avogadro’s number. So, ~2.83 × 10^6^ of the CL7-sfGFP fusion protein molecules per cell were displayed by *P. pastoris* cells using this method. Of note, ~17.8 mg CL7-sfGFP per 1 g dry yeast cell was immobilized on the yeast surface according to the calculation in the following. For each assay, about 1.5 mL of yeast cells was collected and weighted (~51.9 mg). They were then resuspended and diluted in 1 mL TBS following measurement of their OD_600_ by UV/VIS. The calculated OD_600_ of 51.9 mg yeast cells was 23 ± 0.2, therefore the OD_600_ of 1 g wet yeast cells was 443.2. The mass of dehydrated yeast cells from 1 mL wet yeast cells was 10.8 ± 0.1 mg. Thus, the mass ratio between dry and wet *P. pastoris* cells is 26.2%. Finally, the CL7-sfGFP immobilized by 1 g dry yeast cells (m) was calculated according to the formula: m = m1*n/R (m1, 10.4 μg, the mass cof CL7-sfGFP immobilized by 1 mL of 1 OD_600_
*P. pastoris* cells; n, 443.2, the OD_600_ of 1 g wet yeast cells; R, 26.2%, the mass ratio between dry and wet *P. pastoris* cells).

### 2.8. Enzyme Activity Assays for the Free and Surface Displayed CL7-huArg I 

The l-ornithine concentration was detected using the colorimetric assay [16], resulting in a calibration curve shown in Figure 10. According to the fluorometric assay results above, we found that ~20.9 μg CL7-huArg I (52.2 kDa) CL7-sfGFP per 1 g dry yeast cell was immobilized on the yeast surface. After enzymatic reactions, the l-ornithine contents in supernatant were analyzed. One unit of enzyme activity is defined as the activity of enzyme that produced 1 μmol of l-ornithine in 1 min. As a result, the activity of free CL7-huArg I is 504.5 U/mg and the activity of displayed enzymes is 261.3 U/mg, indicating that ~50% enzyme activity can be retained after immobilization. The activity of CL7-huArg I that was immobilized by dry yeast cells is 5045 U/g.

## 3. Materials and Methods

### 3.1. Plasmids and Strains 

*E. coli* strain DH5α was used as the host for DNA manipulation. *E. coli* strain BL21(DE3) was used for protein expression. *P. pastoris* strain GS115 and the vector pPICZαA was obtained from Invitrogen (Carlsbad, CA, USA). The vector pET23a-T [13], pET23a-CL7, pHBM905a-Arg I [17], pCDNA3.1-mCherry and pET23a-sfGFP was constructed and stored in our laboratory. 

### 3.2. Construction of CL7-sfGFP, CL7-mCherry and CL7-huArg I Expression Plasmids 

The expression plasmids for CL7-sfGFP, CL7-mCherry and CL7-huArg I fusion proteins with C-terminal 6× His tags were constructed as follows. The DNA fragments encoding CL7 tag and (G_4_S)_2_ linker were amplified from the plasmid pET23a-CL7 by PCR using the primers CL7F and CL7R. The DNA fragments including (G_4_S)_2_ linker, HRV 3C recognition site and sfGFP were amplified from the plasmid pET23a-sfGFP by PCR using the primers GS3CF, sfGFPF and sfGFPR. The DNA fragment encoding CL7-(G_4_S)_2_-3C-sfGFP was amplified from the above CL7 and sfGFP PCR products using primers CL7F/sfGFPR and cloned into pET23a-T vector employing *Nde* I and *Xho* I. A *Nhe* I site was introduced between HRV 3C recognition site and sfGFP. The plasmid was named pET23a-CL7-sfGFP. Similarly, the DNA fragments encoding mCherry and human Arginase I were amplified from pCDNA3.1-mCherry and pHBM905a-Arg I [17]. The plasmids were constructed and named pET23a-CL7-mCherry and pET23a-CL7-huArg I using the primer pairs mCherryF/mCherryR and huArgIF/huArgIR. All the primers used are shown in Table 3.

### 3.3. Construction of the P. pastoris Surface Display Plasmids 

The plasmids for *P. pastoris* surface display of SED1 fusion proteins with a N-terminal HA tag were constructed as follows. The gene encoding SED1 protein [12] without original secretion signal sequence was obtained by PCR from the genome of *S. cerevisiae* strain INVSC1 using the primers SED1F/SED1R. The above PCR products were used as templates for obtaining gene encoding HA-SED1 using the primers ZαHAF, HA-MCSF, MCS-GSF and SED1R. The restriction enzyme sites including *Nhe* I, *Nde* I, *Hpa* I, *Apa* I and a glycine-serine peptide linker were inserted between HA and SED1. The corresponding DNA fragment was cloned into the vector pPICZαA between *EcoR* I and *Sac* II, producing the plasmid termed pPICZαA-HA-SED1. The *E. coli* Im7 gene was obtained by PCR from pPICZαA-Im7 using the primers Im7F and Im7R. The corresponding DNA fragment was cloned into pPICZαA-SED1 between *Nhe* I and *Apa* I, producing the plasmid called pPICZαA-HA-Im7-SED1. The primers used are shown in Table 4.

### 3.4. Expression and Purification of CL7-mCherry, CL7-sfGFP and CL7-huArg I

The plasmids pET23a-CL7-mCherry, pET23a-CL7-sfGFP and pET23a-CL7-huArg I were transformed into BL21(DE3) chemical competent cells and plated on Luria Bertani broth (LB) plates containing 100 μg/mL ampicillin. Then, one single colony of BL21(DE3) cells on the plates was grown overnight at 37 °C and 200 rpm in 250 mL flasks containing 50 mL of Luria Bertani broth (LB) containing 100 μg/mL ampicillin. The precultured cells were used to inoculate 1 L flasks containing 100 mL of Luria Bertani broth (LB) with 100 μg/mL ampicillin and were grown at 37 °C at 200 rpm. When OD_600_ reached 0.6, the cells were induced with 1 mM IPTG, grown for another 4 h and then harvested by centrifugation. 

The bacterial pellet corresponding to 100 mL of culture was washed two times in Tris-buffered saline (TBS, 50 mM NaCl, 50 mM Tris-HCl, pH 7.6) and resuspended in 25 mL of TBS buffer containing 1 mM PMSF and 10 mM imidazole. The solution was sonicated for 10 min and centrifuged at 14,000 rpm at 4 °C for 0.5 h. The supernatant was used for Ni^2+^-affinity chromatography. The reins binding with fusion proteins were washed with 25mL of 20 mM imidazole in TBS five times and were eluted with 4 mL of 50 mM, 100 mM, 150 mM and 200 mM imidazole in TBS sequentially. The protein samples were fractionated by 12% SDS-PAGE gels. The bands of CL7 fusion protein in Figure 3 were analyzed by ImageJ (National Institutes of Health, Bethesda, MD, USA). The recovery efficiency of CL7 fusion protein eluted with 4 mL of 50, 100, 150 and 200 mM imidazole were calculated (Table 1). The CL7-sfGFP and CL7-mCherry fusion proteins eluted with 200 mM imidazole and CL7-huArginase I eluted with 150 mM and 200 mM imidazole were collected, concentrated and determined by the Bradford method using a protein assay kit from Beyotime (Beijing, China). 

### 3.5. Yeast Transformation, Cultivation and Induction

The plasmids pPICZαA-HA-Im7-SED1 and pPICZαA-HA-SED1 were digested with *Pme* I and transformed into the GS115 electrocompetent cells according to the manual. Transformants were isolated by incubation at 28 °C for 48 h on YPD plates supplemented with 100 μg/mL of Zeocin. Then, five to 10 single colonies of transformants were inoculated in 20 mL of BMGY in 250 mL flasks and cultivated at 28 °C under 200 rpm. After 24 h, the cells were centrifuged at 5000× *g* for 5 min, resuspended in 20 mL of BMMY medium containing 1% (*v*/*v*) methanol and continued to grow at 28 °C, 200 rpm for 24 h. 

### 3.6. Flow Cytometry 

For flow cytometry analysis [12], the *P. pastoris* cells were centrifuged at 3000× *g* for 1 min. The collected cells were washed twice by ice-cold water, resuspended and blocked in 1 mL of phosphate buffered saline (PBS, 137 mM NaCl, 10mM Na_2_HPO_4_·12 H_2_O, 2.7 mM KCl, 2 mM KH_2_PO_4_, pH 7.4) with 1 mg/mL BSA for 1 h at 4 °C with rotation. Then, 1 µL of mouse anti-HA tag monoclonal antibodies and 5 µg of CL7-sfGFP proteins were added to the cell suspension of 1000 µL and were incubated at room temperature with rotation for 2 h. The cells were then washed three times with PBS and resuspended in 200 µL of PBS with the addition of 1 µL of DyLight 649-conjugated goat anti-mouse IgG(H+L) antibodies and were then incubated at room temperature with rotation for 1 h. At last, the cells were washed three times with PBS and suspended in 1mL of PBS. The yeast cells were examined using a flow cytometer (CytoFLEX, Beckman Coulter, Suzhou, China; Excitation wavelength, 488 nm and 638 nm; Emission wavelength, 525 nm and 660 nm) to estimate the percentage of the fluorescence positive cells. 

### 3.7. Fluorescence Microscopy

For fluorescence microscopy analysis [12], cells of GS115/pPICZαA, GS115/pPICZαA HA-SED1 and GS115/pPICZαA HA-Im7-SED1 were washed and blocked as described above. The cells were resuspended in 1 mL of PBS and 1 µL of mouse anti-HA tag monoclonal antibodies as well as 5 µg of CL7-mCherry protein were added and incubated at room temperature with rotation for 2 h. The cells were washed three times with PBS and resuspended in 200 µL of PBS with the addition of 1 µL of FITC–conjugated goat anti-mouse IgG(H+L) antibodies and were then incubated at room temperature for 1 h with rotation. Finally, the cells were washed three times with PBS, resuspended in 1 mL of PBS and examined by a fluorescence microscopy (IX73, Olympus, Tokyo, Japan).

### 3.8. Blue Light Transmitter Analysis 

For direct visualization of the yeast cell surface immobilized CL7-sfGFP proteins, a blue light transmitter with the wavelength from 440–485 nm (Sangon Biotech, Shanghai, China) was used. The GS115/pPICZαA HA-SED1 and GS115/pPICZαA HA-Im7-SED1 cells were washed and blocked as described above. The cells were resuspended in 1 mL of PBS and 10 µg of CL7-sfGFP proteins were added to each cell suspension and were incubated at room temperature with rotation for 1 h. The cells were then washed five times with PBS and analyzed under the blue light transmitter. 

### 3.9. Western Blotting Analysis

For Western blotting analysis [10], the cells of GS115/pPICZαA, GS115/pPICZαA HA-SED1 and GS115/pPICZαA HA-Im7-SED1 were washed twice by distilled water, adjusted to OD_600_ = 10 and boiled in SDS-PAGE buffer (10% SDS, 50% glycerol, 5% 2-mercaptoethanol, 0.5% bromophenol blue, 0.25 M Tris-HCl, pH6.8). The protein samples were fractionated by an 8% SDS-PAGE gel and electro-transferred onto a PVDF membrane. The PVDF membrane was blocked with 5% skimmed milk in Tris-buffered Saline Buffer containing 0.5% Tween 20 (TBST, 50 mM NaCl, 50 mM Tris-HCl, 0.5% Tween 20, pH 7.6) and incubated with mouse anti-HA monoclonal antibody (1:2000) in 5% skimmed milk in TBST for 1 h at room temperature and washed five times. The PVDF membrane was then incubated with HRP-conjugate goat anti-mouse IgG(H+L) antibody (1:2000) in 5% skimmed milk in TBST for 1 h at room temperature. The PVDF membrane was washed five times and the protein bands were visualized using a chemiluminescence detection reagent (Millipore Corporation, Billerica, MA, USA). 

### 3.10. Fluorometric Assay

For the quantification of yeast surface indirectly displayed CL7-sfGFP fusion proteins, the GS115/pPICZαA-HA-SED1 and GS115/pPICZαA-HA-Im7-SED1 cells were washed and blocked as described above. The fluorometric assay was performed as reported by Shibasaki et al. with some modifications [18], using the fluorescence spectrophotometer F-4600 (HITACHI, Tokyo, Japan). The excitation wavelength and emission wavelength of the CL7-sfGFP fusion proteins were determined by wavelength scanning. Then, 490 nm and 512 nm were chosen as the optimal excitation and emission wavelength, respectively. The fluorescence intensities of 0 µg, 1 µg, 2 µg, 3 µg, 4 µg, 5 µg, 6 µg, 7 µg, 8 µg, 9 µg and 10 µg of CL7-sfGFP fusion proteins in 1 mL of TBS were measured. The calibration curve was prepared with the fluorescence intensities of 4 µg to 10 µg of CL7-sfGFP fusion proteins (Figure 8).

Different amounts of CL7-sfGFP fusion proteins were added into 0.5 mL of the GS115/pPICZαA-HA-Im7-SED1 cells (Table 1). Then, 120 μg of CL7-sfGFP fusion proteins were also added into 0.5 mL of the GS115/pPICZαA-HA-SED1 cells. These cells were incubated at room temperature with rotation for 1 h and were then washed five times with TBS and resuspended in 1 mL of TBS. The GS115/pPICZαA-HA-Im7-SED1 and GS115/pPICZαA-HA-SED1 cells were diluted. Their fluorescence intensities and OD_600_ were examined and shown in Table 1. In addition, the fluorescence intensity/OD_600_ (F/A) values were analyzed with GraphPad Prism 5 and are given in Figure 9. 

### 3.11. Human Arginase I Activity Assay 

The bio-conversion of l-ornithine from l-Arginine by *P. pastoris* surface indirectly displayed human Arginase I was examined [16,17]. In a typical experiment, 0.5 mL of GS115/pPICZαA HA-Im7-SED1 cells of 20 OD_600_ displaying HA-Im7-SED1 fusion proteins was washed five times with bicarbonate buffer (100 mM, pH = 10). Excessive amounts of CL7-huArg I fusion proteins were added into each cell suspension and the cell suspensions were incubated with rotation at room temperature for 2 h. The cells were washed five times with bicarbonate buffer (100 mM, pH = 10) and resuspended in 0.5 mL of bicarbonate buffer (100 mM, pH = 10). The enzyme activity was examined at 40 °C for 10 min in 1 mL of reaction volume (bicarbonate buffer, 50 mM, pH 10; 1 mM MnCl_2_; 40 mM l-Arginine). The reactions were stopped by boiling at 100 °C for 5 min and the reaction solutions were centrifuged at 5000g for 2 min. Then, 100 μL of 1M TCA was added to 900 μL of the reaction supernatant and the concentration of l-ornithine was detected using the Chinard colorimetric assay [16]. 

## 4. Conclusions and Perspective 

In conclusion, we established a novel highly efficient indirect *P. pastoris* surface display system using the ultra-high-affinity interaction between Im7 and CL7 in this study. The Im7-SED1 fusion proteins were successfully expressed on the yeast surfaces. The interaction of surface displayed Im7 proteins with CL7 fusion proteins was confirmed by the blue light transmitter, flow cytometry and fluorescence microscopy. Generally, ~2.8 × 10^6^ CL7-sfGFP fusion protein molecules per cell were displayed indirectly by *P. pastoris*, which was examined and calculated by fluorometric assays. The bio-conversion of l-ornithine from l-Arginine by free and *P. pastoris* surface displayed human Arginase I was examined, indicating that the enzymes can be efficiently displayed on the cell surface by this method. In future, this efficient surface display system can be further applied as a robust platform for the development of vaccine, bio-sensor, electrochemical devices and bio-catalysts.

## Figures and Tables

**Figure 1 molecules-24-01483-f001:**
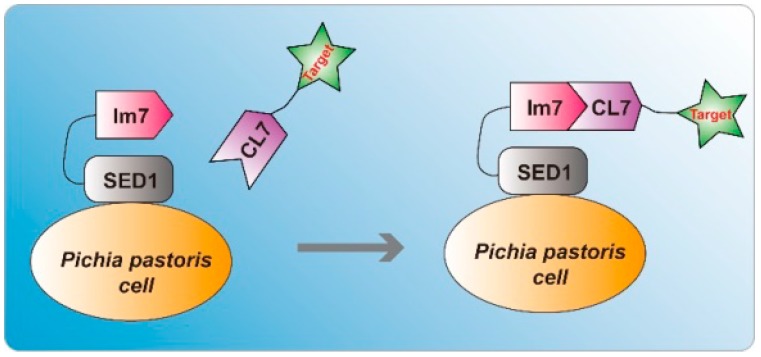
Schematic illustration of the indirect *P. pastoris* surface display system through the ultra-high-affinity interaction between Im7 and CL7. The Im7-SED1 fusion proteins are expressed and displayed on the surface of *P. pastoris* cells. The addition of CL7 tagged proteins leads to the indirect display of target proteins.

**Figure 2 molecules-24-01483-f002:**
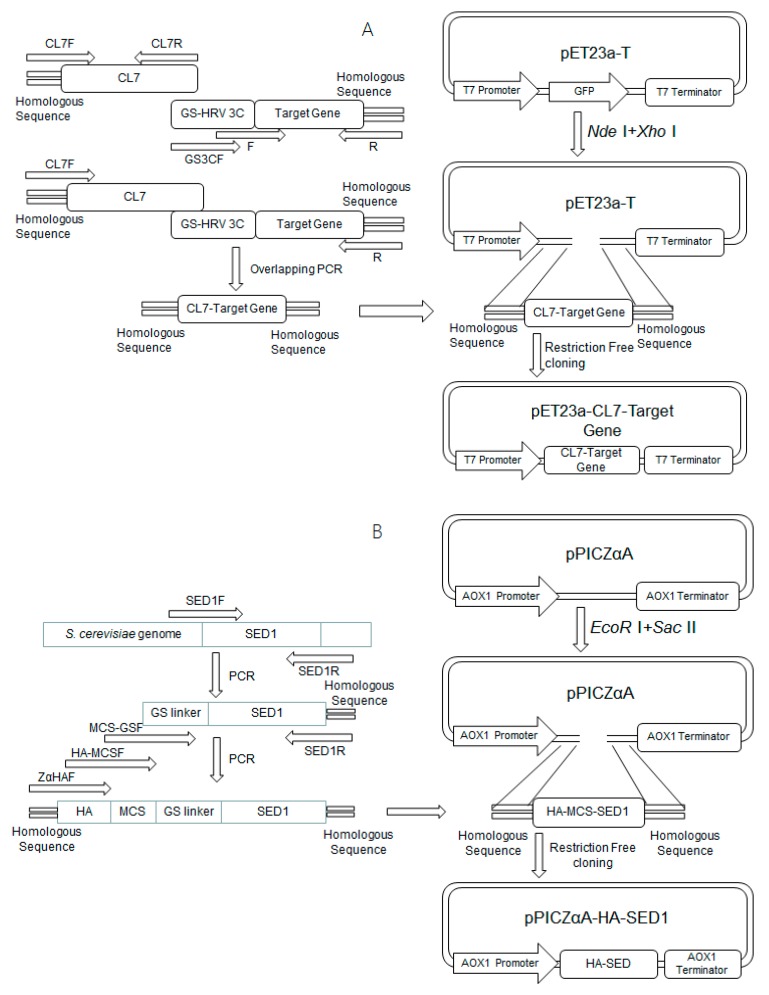
Schemes for the construction of plasmids including (**A**) pET23a-CL7-sfGFP, pET23a-CL7-mCherry, pET23a-CL7-huArg I; (**B**) pPICZαA-HA-SED1; and (**C**) pPICZαA-HA-Im7-SED1.

**Figure 3 molecules-24-01483-f003:**
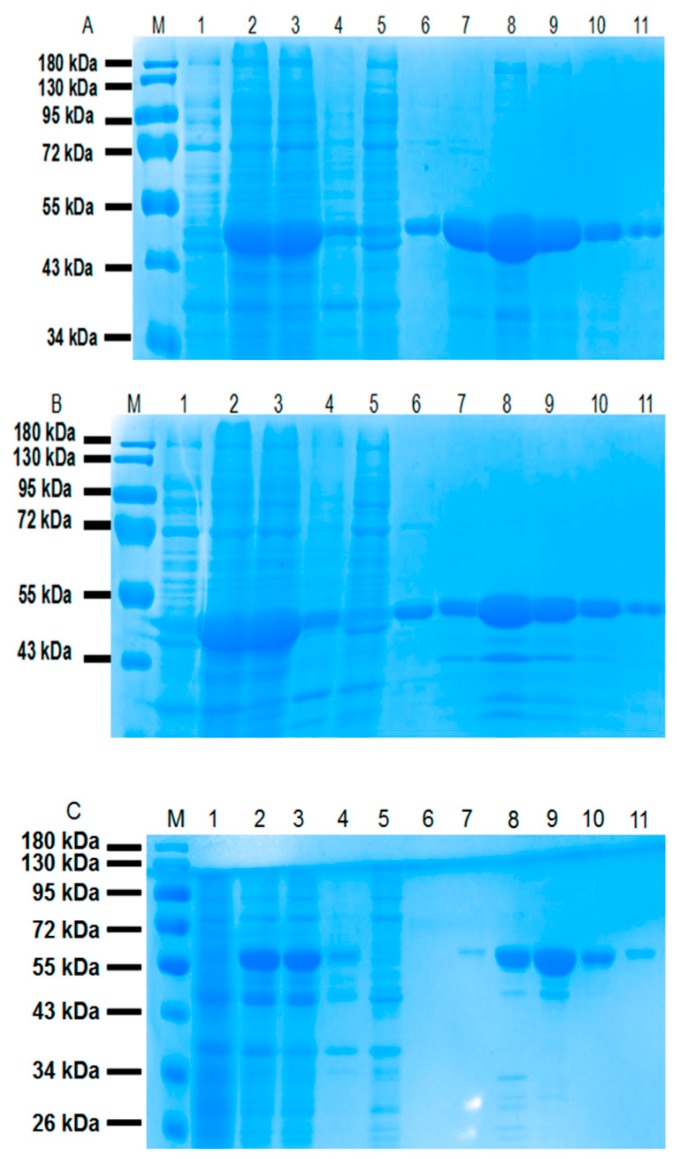
SDS-PAGE analysis of the (**A**) CL7-sfGFP, (**B**) CL7-mCherry and (**C**) CL7-huArg I fusion proteins purified by Ni-NTA. M: protein ladder (Thermo Scientific, Product# 26616); Lane 1: *E. coli* cells without IPTG (Isopropyl β-d-Thiogalactoside); Lane 2: *E. coli* cells induced with 1 mM IPTG; Lane 3: supernatant after sonication; Lane 4: debris after centrifugation; Lane 5: flow-through of supernatant after binding with Ni^2+^-resin; Lane 6: flow-through of Ni^2+^-resin washed with 20 mM imidazole; Lane 7 to Lane 10: elution of 4mL TBS (tris-buffered saline) containing 50 mM, 100 mM, 150 mM and 200 mM imidazole, respectively; Lane 11: the resin resuspended in 4 mL TBS.

**Figure 4 molecules-24-01483-f004:**
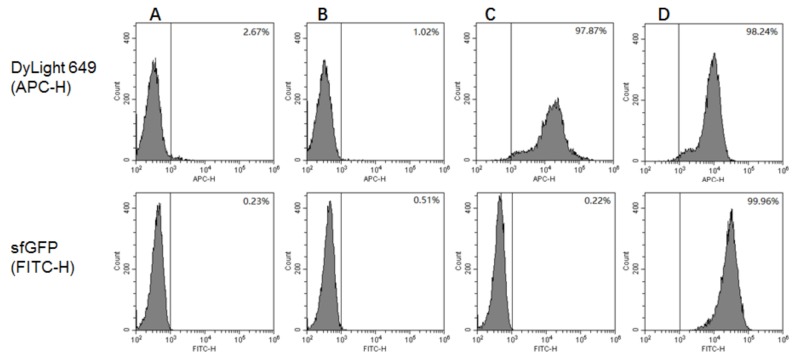
Flow cytometry analysis of *P. pastoris* cells including (**A**) unlabeled GS115/pPICZαA, (**B**) GS115/pPICZαA, (**C**) GS115/pPICZαA HA-SED1 and (**D**) GS115/pPICZαA HA-Im7-SED1. All the cells except (**A**) were treated with DyLight 649-conjugated goat anti-mouse IgG(H+L) antibodies (top line) and with CL7-sfGFP and mouse anti-HA tag monoclonal antibodies (bottom line).

**Figure 5 molecules-24-01483-f005:**
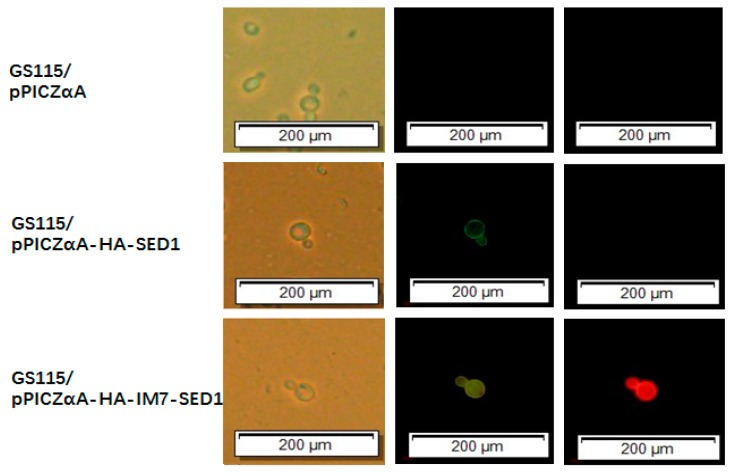
Fluorescence microscopy analysis of *P. pastoris* yeast cells including GS115/pPICZαA (first line), GS115/pPICZαA HA-SED1 (second line) and GS115/pPICZαA HA-Im7-SED1 (third line). All the cells were treated with (middle row) mouse anti-HA tag monoclonal antibodies together with FITC (fluorescein isothiocyanate)-conjugated goat anti-mouse IgG (H+L) antibodies and with CL7-mCherry (right row).

**Figure 6 molecules-24-01483-f006:**
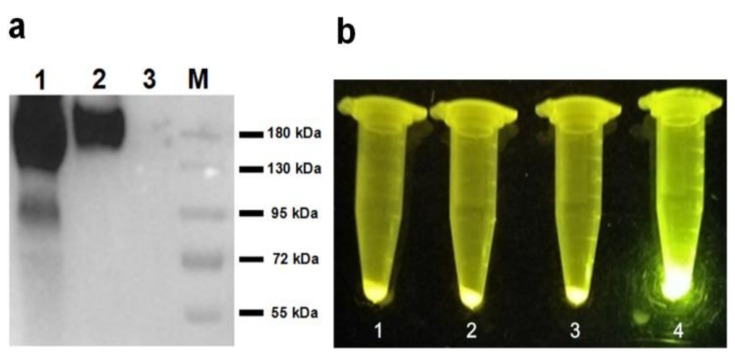
(**a**) The western-blot analysis indicated that the HA-Im7-SED1 and HA-SED1 fusion proteins were displayed on the surface of *P. pastoris* cells. (**b**) The blue light transmitter analysis showed that CL7-sfGFP proteins were indirectly immobilized on the surface of *P. pastoris* cells.

**Figure 7 molecules-24-01483-f007:**
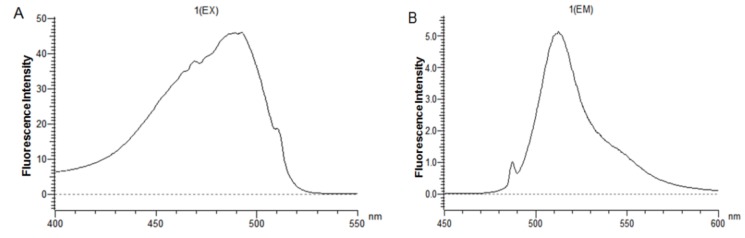
Scanning the (**A**) excitation wavelength and (**B**) emission wavelength of CL7-sfGFP in TBS buffer (pH 7.6).

**Figure 8 molecules-24-01483-f008:**
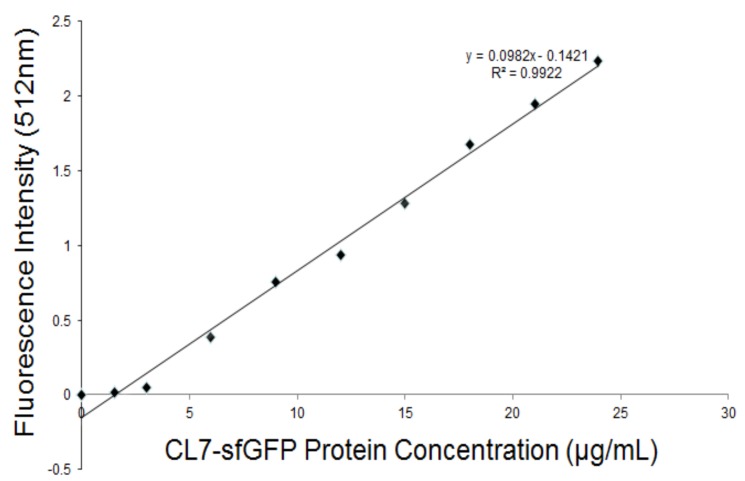
The calibration curve and linear regression equation of the CL7-sfGFP fusion proteins in TBS buffer (pH 7.6).

**Figure 9 molecules-24-01483-f009:**
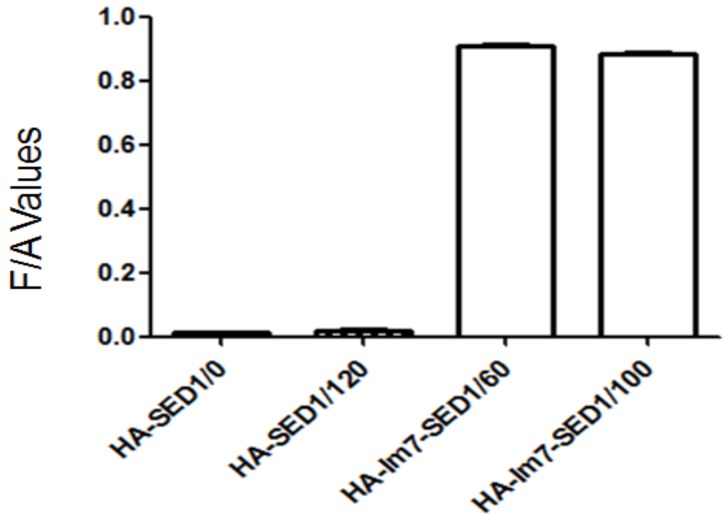
The fluorescence intensity/OD600 (F/A) values of yeast cells incubated with different amounts (μg) of CL7-sfGFP.

**Figure 10 molecules-24-01483-f010:**
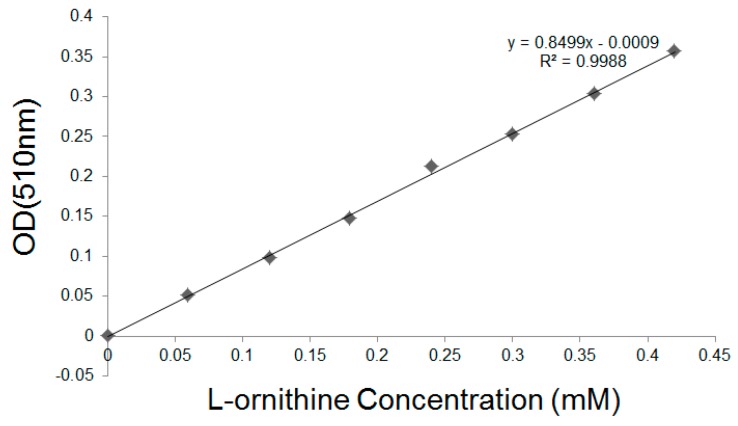
The calibration curve of l-ornithine.

**Table 1 molecules-24-01483-t001:** Recovery efficiency of CL7 fusion protein purified by Ni^2+^-affinity chromatography.

Protein	Supernatant	50 mM	100 mM	150 mM	200 mM
CL7-sfGFP	100%	11.4%	20.9%	12.0%	6.3%
CL7-mCherry	100%	4.1%	12.9%	7.6%	5.3%
CL7-huArg I	100%	*	10.8%	19.7%	8.4%

* The band intensity was too low to be detected.

**Table 2 molecules-24-01483-t002:** The fluorescence intensity/OD_600_ (F/A) of sf-GFP on the yeast surface.

Cells	GS115/pPICZα-HA-SED1		GS115/pPICZα-HA-Im7-SED1	
Protein Quantity (μg)	0	120	60	100
Fluorescence Intensity/OD_600_	0.012 (± 0.0039)	0.017 (± 0.0054)	0.91 (± 0.063)	0.88 (± 0.075)

**Table 3 molecules-24-01483-t003:** The primers used for the construction of plasmids including pET23a-CL7-sfGFP, pET23a-CL7-mCherry and pET23a-CL7-huArg I.

Primer	Sequence (5′→3′)
CL7F	gaaggagatatacatatgagcaaaagcaatgaaccgggtaaag
CL7R	tccaccacctgaaccacctcctccgccttcaatatcaatgttgcgtttcg
GS3CF	ggaggaggtggttcaggtggtggaggcagtttggaggttttgttccagggtccagctag
sfGFPF	gaggttttgttccagggtccagctagcgtgagcaagggcgaggagctgttc
sfGFPR	ggtggtggtggtggtgctcgagttccttgtacagctcgtccatgcc
mCherryF	gaggttttgttccagggtccagctagcatggttagcaaaggggaggaggataac
mCherryR	ggtggtggtggtggtgctcgagttccttgtacagctcgtccataccgc
huArgIF	gaggttttgttccagggtccagctagcagtgctaagtccagaacgattggtattattg3′
huArgIR	ggtggtggtggtggtgctcgagctttggtgggttcaaatagtcaattggt

**Table 4 molecules-24-01483-t004:** The primers used for the construction of plasmids pPICZαA-HA-SED1 and pPICZαA-HA-Im7-SED1.

Primer	Sequence (5′→3′)
SED1F	aggcggtagcggaggcggagggtcgcaattttccaacagtacatctgcttcttcc
SED1R	gaaagctggcggccgccgcggtcattataagaataacatagcaacaccagccaaac
ZαHAF	aaagagaggctgaagctgaattctacccatacgacgttccagactacgctggaggctct
HA-MCSF	cgttccagactacgctggaggctctgctagccatatggttaacgggcc
MCS-GSF	tgctagccatatggttaacgggcccggaggcggtagcggaggcggagggtc
Im7F	ctacgctggaggctctgctagcatggaattgaagaactccatctccgact
Im7R	ctccgctaccgcctccgggcccaccttgtttaaaacctggcttaccgttg

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
