# Peer review of "A Highly Efficient Indirect P. pastoris Surface Display Method Based on the CL7/Im7 Ultra-High-Affinity System"

_molecules, 2019, doi:10.3390/molecules24081483_

Round 1

Reviewer 1 Report

This work showed a novel indirect yeast surface display system based on the high-affinity complex of E. coli Im7 and CL7 tag. Im7 was fused to the N-terminus of SED1 protein from S. cerevisiae and the fusion protein was displayed on the cell surface of P. pastoris. Using the CL7-sfGFP and CL7-mCherry fusion proteins, authors demonstrated the interaction of the Im7 and the CL7 fusion proteins. Finally, the bioconversion of L-ornithine from L-arginine by human arginase I immobilized on P. pastoris cell surface was studied.

While the science described in this paper appears sound, the manuscript itself requires revision before it is ready to publish.

How much enzyme (e.g. mg protein per g cell) could be immobilized on the cell surface using this adsorption system?

How much enzyme activity was retained by arginase after immobilization?

Add molecular size of CL7 tag and Im7 in the introduction section.

Figure 4, 7 and 8. Increase figure size and quality.

Author Response

RIVEWER 1:

This work showed a novel indirect yeast surface display system based on the high-affinity complex of E. coli Im7 and CL7 tag. Im7 was fused to the N-terminus of SED1 protein from S. cerevisiae and the fusion protein was displayed on the cell surface of P. pastoris. Using the CL7-sfGFP and CL7-mCherry fusion proteins, authors demonstrated the interaction of the Im7 and the CL7 fusion proteins. Finally, the bioconversion of L-ornithine from L-arginine by human arginase I immobilized on P. pastoris cell surface was studied.

While the science described in this paper appears sound, the manuscript itself requires revision before it is ready to publish.

How much enzyme (e.g. mg protein per g cell) could be immobilized on the cell surface using this adsorption system? How much enzyme activity was retained by arginase after immobilization?

Response: According to the calculation described in section 2.8 and 2.9, ~20.9 mg enzyme per 1 g dry cell was immobilized on the yeast surface, while ~ 50% enzyme activity was retained regarding human Arginase I after immobilization.

Add molecular size of CL7 tag and Im7 in the introduction section.

Response: Thanks for your suggestion. We added the size information of CL7 tag and IM7 in the introduction section accordingly.

Figure 4, 7 and 8. Increase figure size and quality.

Response: Thanks for your suggestion and we supplied new Figures accordingly.

Reviewer 2 Report

This paper is dealing with construction of a display yeast system for immobilization of proteins.

In general, Introduction section is well documented.

In the paper, some typo errors should corrected. Delete 3´ in one primer of Table 3. Change "reins" by "resin" in line 291. Change In lines 221, 227, 230, 372, 373, 380, 394 and 395 change "argine" by "arginine". Change in Line 156 "incubate" by "incubated". Change in line 386 “coclusion” by “conclusion”.

Results and discussion. In general this section can be improved. Some data treatment should be re-evaluated. Some additional experimental information should be added.

Section of Plasmid construction can be improved. Add a figure with a scheme of plasmid construction adding a cloning workflow indicating steps and drawing plasmid with promoters, restriction sites, etc.

In the CL7 fusion expression section, IMAC-Chromatography proves protein display, but I think authors should add a table with protein recovery of every step, and not only SDS-PAGE.

If well flow cytometry analysis demonstrates protein display in Pichia pastoris, cell counts are in different magnitudes. I think authors should use the same maximum magnitude of counts in axis Y.

Authors suggests some glycosylation because increasing in molecular weights of protein determined by western blotting. I think this should be experimentally proved or to estimate based on amino acid structure of fused proteins. Indicate MW increasing in percentage or Da.

In Fluorescence spectra of Figure 7, add units in axis Y. In Figure 8, can be observed that detection limit is 4 micrograms per mL. Why there is no linearity at lower concentrations? Some explanation for this?

Authors propose an F/A value for determination of immobilization of fused protein on Pichia pastoris cells, determining fluorescence intensity/OD cells ratio. I suggest to authors, based on determination of mass of immobilized proteins, try to determine this ratio with dry weight cells. Figure 10 and Table 2 contain data from enzyme activity of arginase. I think absorbance units so low (<0.1) can produce some determination errors in concentration of L-ornithine. In addition, I do not know, how authors can measure absorbance higher than 4 (see data of OD at 510 nm for L-ornithine). This data is for diluted concentrations? Clarify this or modify this data from the table. The same comments for quantification of cell concentration at 600 nm.

I think authors should include an acknowledgments section.

Author Response

RIVEWER 2:

This paper is dealing with construction of a display yeast system for immobilization of proteins.

In general, Introduction section is well documented.

In the paper, some typo errors should be corrected. Delete 3´ in one primer of Table 3. Change "reins" by "resin" in line 291. Change In lines 221, 227, 230, 372, 373, 380, 394 and 395 change "argine" by "arginine". Change in Line 156 "incubate" by "incubated". Change in line 386 “coclusion” by “conclusion”.

Response: Thanks a lot for help us find typos and we have made changes accordingly. We rewrote the whole paper and polished language at this time.

Results and discussion. In general this section can be improved. Some data treatment should be re-evaluated. Some additional experimental information should be added.

Response: Thanks. We rewrote this section in the revised version.

Section of Plasmid construction can be improved. Add a figure with a scheme of plasmid construction adding a cloning workflow indicating steps and drawing plasmid with promoters, restriction sites, etc.

Response: Thanks for your suggestion and we supplied new Figure 2 to demonstrate the workflow of plasmid construction and the plasmids’ information

In the CL7 fusion expression section, IMAC-Chromatography proves protein display, but I think authors should add a table with protein recovery of every step, and not only SDS-PAGE.

Response: Thanks. We added a new Table 1 which contains the information you required.

If well flow cytometry analysis demonstrates protein display in Pichia pastoris, cell counts are in different magnitudes. I think authors should use the same maximum magnitude of counts in axis Y.

Response: Thanks. We now showed the same Y axis in Figure 4 according to your suggestion.

Authors suggests some glycosylation because increasing in molecular weights of protein determined by western blotting. I think this should be experimentally proved or to estimate based on amino acid structure of fused proteins. Indicate MW increasing in percentage or Da.

Response: We agreed with you. According to sequence analysis in http://www.cbs.dtu.dk/

services/NetNGlyc/#opennewwindow, SED1 has seven N-glycosylation sites while Im7 has no N-glycosylation sites. Therefore, the SED1 fusion proteins were highly possible in glycosylated forms. As shown in Figure 7, the molecular weight of HA-SED1 and HA-Im7-SED1 fusion proteins were both around 180 kDa, which are 140 kDa larger than their theoretical MW (HA-SED1, 35.6 kDa; HA-Im7-SED1, 45 kDa).

In Fluorescence spectra of Figure 7, add units in axis Y. In Figure 8, can be observed that detection limit is 4 micrograms per mL. Why there is no linearity at lower concentrations? Some explanation for this?

Response: In fluorescence spectra of Figure 8 (original Figure 7), the units of axis Y were relative Fluorescence Intensity. We repeated the fluorescence experiments and re-measured the fluorescence of 0 µg, 1.5 µg, 3 µg, 6 µg, 9 µg, 12 µg, 15 µg, 18 µg, 21 µg, 24 µg of CL7-sfGFP fusion proteins in 1 ml of TBS (pH 7.6). The results show that there was no linearity when protein concentrations were lower than 6 µg, possibly due to the detection limits of fluorescence spectrum or the system error.

Authors propose an F/A value for determination of immobilization of fused protein on Pichia pastoris cells, determining fluorescence intensity/OD cells ratio. I suggest to authors, based on determination of mass of immobilized proteins, try to determine this ratio with dry weight cells.

Response: Thanks for your kind suggestion. We recalculated the immobilized proteins on the yeast surface, resulting in ~ 17.8 mg sfGFP and ~ 20.9 mg human Arginase I per 1 g dry weight cells.

Figure 10 and Table 2 contain data from enzyme activity of arginase. I think absorbance units so low (<0.1) can produce some determination errors in concentration of L-ornithine.

Response: Thanks for your suggestion. We re-measured the calibration curve of L-ornithine and showed the results in new Figure 10.

In addition, I do not know, how authors can measure absorbance higher than 4 (see data of OD at 510 nm for L-ornithine). This data is for diluted concentrations? Clarify this or modify this data from the table. The same comments for quantification of cell concentration at 600 nm.

Response: Thanks a lot. The reaction solutions were diluted and the concentrations of L-ornithine were measured. According to your suggestion, the data was modified in the table. In addition, the same cell concentration at 600 nm were modified.

I think authors should include an acknowledgments section.

Response: We added a new acknowledgements section as you required.